# Structure Prediction and Analysis of Hepatitis E Virus Non-Structural Proteins from the Replication and Transcription Machinery by AlphaFold2

**DOI:** 10.3390/v14071537

**Published:** 2022-07-14

**Authors:** Adeline Goulet, Christian Cambillau, Alain Roussel, Isabelle Imbert

**Affiliations:** 1Aix-Marseille Université, Centre National de la Recherche Scientifique, UMR 7255, LISM, 31 Chemin Joseph Aiguier, 13009 Marseille, France; adeline.goulet@univ-amu.fr (A.G.); alain.roussel@univ-amu.fr (A.R.); 2School of Microbiology, University College Cork, T12 YT20 Cork, Ireland; cambillau.alphagraphix@gmail.com; 3AlphaGraphix, 24 Carrer d’Amont, 66210 Formiguères, France

**Keywords:** Hepatitis E virus, nonstructural proteins, viral replication/transcription enzymes, AlphaFold2, macro domain, helicase, RNA-dependent RNA polymerase

## Abstract

Hepatitis E virus (HEV) is a major cause of acute viral hepatitis in humans globally. Considered for a long while a public health issue only in developing countries, the HEV infection is now a global public health concern. Most human infections are caused by the HEV genotypes 1, 2, 3 and 4 (HEV-1 to HEV-4). Although HEV-3 and HEV-4 can evolve to chronicity in immunocompromised patients, HEV-1 and HEV-2 lead to self-limited infections. HEV has a positive-sense single-stranded RNA genome of ~7.2 kb that is translated into a large pORF1 replicative polyprotein, essential for the viral RNA genome replication and transcription. Unfortunately, the composition and structure of these replicases are still unknown. The recent release of the powerful machine-learning protein structure prediction software AlphaFold2 (AF2) allows us to accurately predict the structure of proteins and their complexes. Here, we used AF2 with the replicase encoded by the polyprotein pORF1 of the human-infecting HEV-3. The boundaries and structures reveal five domains or nonstructural proteins (nsPs): the methyltransferase, Zn-binding domain, macro, helicase, and RNA-dependent RNA polymerase, reliably predicted. Their substrate-binding sites are similar to those observed experimentally for other related viral proteins. Precisely knowing enzyme boundaries and structures is highly valuable to recombinantly produce stable and active proteins and perform structural, functional and inhibition studies.

## 1. Introduction

Hepatitis E virus (HEV) is a major cause of acute self-limiting viral hepatitis in humans globally and has infected one-third of the world’s population [1]. At least 20 million new infectious occur annually, resulting in ~60,000 deaths. Although the lethality rate is less than 3% in the general population, it can reach up to 30% in infected pregnant women in Southeast Asia [2]. Considered for a long while a public health issue only in developing countries, HEV infection is now a global public health concern. In developed countries, HEV has become one of the most successful zoonotic viral diseases [3]. Interestingly, unlike hepatitis B and C viruses, which have only a hepatic tropism, HEV infection is associated with multiple extrahepatic diseases, including neurological diseases, in 10% of cases.

HEV is the only member of the *Hepeviridae* family, subfamily *Orthohepevirinae*. This subfamily comprises 4 genera including the genus *Paslahepevirus*. Most human infections are with the specie *Paslahepevirus balayani* genotypes 1, 2, 3, and 4 and less frequently 7 [4]. Among the 4 major HEV genotypes infecting humans, a clear epidemiology dichotomy is observed between developing and industrialized countries: the HEV genotypes 1 (HEV-1) and HEV-2 are restricted to humans and mostly affect Asia and Africa. They are transmitted through faecally contaminated drinking water and are responsible for acute hepatitis large-scale epidemics [5]. Conversely, HEV-3 and HEV-4 are common in industrialized nations and are transmitted by consumption of animal raw meat products, organ transplantation or blood transfusions. They do not cause severe disease in pregnant women in contrast to HEV-1 and HEV-2. Moreover, although HEV-1 and HEV-2 strains usually lead to self-limited infections, HEV-3 and HEV-4 can evolve to chronicity in immunocompromised patients.

Although the viral positive-sense single-stranded RNA genome of ~7.2 kb was sequenced in 1990 [6], HEV remains an enigmatic virus regarding viral life cycle and replication machinery. Originally considered as a non-enveloped virus, it was shown in 2016 that HEV shed from infected cells as a quasi-enveloped virus (named eHEV) [7]. Thus, HEV is non-enveloped in faeces, whereas viral particles released from the basal membrane of hepatocytes enter the serum coated with a lipid envelope. Upon entry into the host cell, the viral RNA genome expression starts with the cap-dependent translation of the unique open reading frame (ORF1), located in the 5′ two-thirds of the genome, and that is directly accessible to cellular ribosomes. This leads to the production of a large replicative polyprotein of 1644 residues for HEV-1 and of 1708 residues for HEV-3. A unique computational homology analysis of this polyprotein, conducted by Koonin et al. 30 years ago, has predicted 7 functional domains, all essential for efficient viral replication [8,9]: the methyltransferase (MTase), Y, putative papain-like protease (PLP), proline-rich hinge, macro, RNA helicase and the RNA-dependent RNA-polymerase (RdRp) domains. Traditionally, the polyprotein encoding the nonstructural proteins (nsPs) of mammalian (+)RNA viruses is processed by protease(s) to release individual and functional proteins. Surprisingly, whether the pORF1 polyprotein is processed into subunits or not [10,11,12,13,14,15,16,17,18,19,20,21,22] remains an open question. Thereby, this knowledge gap on the composition and structure of the HEV replication complex(es) prevents the development of antivirals. Regardless of the polyprotein pORF1 fate, viral replication requires at least the viral RdRp domain. It uses the viral (+)RNA genome as template to replicate the negative-sense RNA strand. The (−)RNA strand serves in turn as template for either the replication of a new (+)RNA genome or for the transcription of the (+) subgenomic (sg) mRNA that contains ORF2 and ORF3. Subsequently, the sg mRNA is translated by ‘leaky scanning’ to produce the pORF2, the viral capsid protein [23] and pORF3. The latter is an ion channel essential for the release of new infectious viral particles [24].

The superfamily concept gathers viruses sharing common features in encoded replicase proteins, genome organization, and replication strategies. In this context, HEV has been classified in the alphavirus-like superfamily, including also brome mosaic virus and *Alphavirus* genus (e.g., Sindbis, Chikungunya, Semliki Forest viruses) [25]. The Alphavirus replication machinery consists of four nonstructural (or replicase) proteins (nsP1-4). Alphavirus nsPs are initially produced as a single polyprotein (named P1234), which is then sequentially processed in a highly-regulated manner [26]. Therefore, it is likely that HEV also uses differential cleavage of its replicative polyprotein to regulate its replication and transcription stages. In addition, this strategy could reconcile the contradictory observations mentioned above.

The recent release of the powerful machine-learning protein structure prediction software AlphaFold2 (AF2) was a revolution in structural biology [27,28,29]. Accurate structure predictions of proteins and their complexes have become possible and easy using AF2 [30]. In particular, it is well adapted to predict structures of long, flexible and multi-domain proteins that cannot be analyzed as a whole using experimental approaches such as X-ray crystallography and electron microscopy. Recently, we applied this method to bacterial viruses (bacteriophages) to determine the structures of their whole host-binding machineries with several multi-domain proteins [31,32]. Moreover, as compared with homology modeling, AF2 produces structures that do not suffer from sequence bias, with an estimate of the prediction reliability for each protein residue given by a confidence score, the predicted local distance difference test (pLDDT, from 0 to 100). Values of pLDDT in the 80–90 range indicate that the structural prediction is comparable to the average resolution (2.5–3.0 Å) experimental structures. Therefore, we reason that AF2 may be a method of choice to predict the boundaries and structures of the HEV replication and transcription complex components.

Here, we applied this approach to the replicase polyprotein pORF1of the human-infecting HEV genotype 3 (HEV-3). The structures of five domains were reliably predicted and showed folds similar to viral methyltransferase, zinc-binding domain of a putative HEV cysteine protease [33], macro, helicase and RdRp. This leads us to propose a rational nomenclature and a structural description of the HEV genotype 3 (named hereafter HEV-3) nonstructural proteins (nsPs), opening, more than 30 years after the HEV identification, new research perspectives to characterize the molecular mechanisms of HEV replication. Due to high sequence similarity between HEV-3 and HEV-1 (~85%), these predictions hold also true for the HEV-1 replicase pORF1 polyprotein.

## 2. Materials and Methods

Structural predictions were conducted on the HEV Kernow-C1 clone (genotype 3; GenBank accession n° HQ389543). We used a Github notebook (https://colab.research.google.com/github/deepmind/alphafold/blob/main/notebooks/AlphaFold.ipynb#scrollTo=XUo6foMQxwS2, accessed on 8 May 2022) to perform the polyprotein segment predictions, as it provides a simple and efficient service. To note, this notebook does not use PDB templates (as do ‘true’ AlphaFold2 servers), thereby providing a totally naive structure prediction. We predicted the structures of two segments, 1–1250 and 1000–1708, as the GPU memory of the notebook’s servers have an upper limit of ~1400 residues. The structures were edited with *Coot* [34] to separate the individual nsPs. Related structures retrieval was performed with Dali [35]. Structure representations were performed with ChimeraX [36]. The polyprotein segments and the individual nsP structures files in PDB format are available as Appendix A. The pLDDT plots were performed with Excel, starting from the B-factors column of the PDB files where they are stored. To note, in AlphaFold2 plots, the pLDDT are represented from red (bad) to blue (good). In Figure 1, with ChimeraX, the pLDDT are represented in an inverted fashion from blue (bad) to red (good).

## 3. Results

### 3.1. AlphaFold2 Structure Prediction of the HEV-3 Polyprotein pORF1

The HEV-3 polyprotein pORF1 of 1708 residues was split into two overlapping segments for AF2 structure predictions, thereby circumventing notebook memory limitations. Segment 1 (residues 1–1250) and segment 2 (residues 1000 to 1708) revealed well-predicted domains, joined together by linkers with low-confidence scores (Figure 1 and Appendix A). Segment 1 exhibits four domains (Figure 1A), which could be ascribed to putative nsP1, nsP2, nsP3 and nsP4, according to their occurrence along the polyprotein chain (Figure 1B). Segment 2 exhibits two domains, the first of which partially overlaps with nsP4 in segment 1 and is followed by nsP5 at the C-terminal end of the polyprotein. Based on the high-confidence scores of the predicted domains, we could identify the boundaries of the five HEV-3 nsPs (Table 1) and analyze each of them.

### 3.2. HEV-3 nsP1: A Putative Capping Pore of the Viral Replication Factory

We determined the nsP1 boundaries at residues 9 and 459 of the polyprotein pORF1 (Table 1). We submitted the predicted structure to the Dali server [41], which reported a significant hit (Z-score of 20.2) with the cryo-EM structure of the Chikungunya virus (CHIKV) nsP1, a membrane-associated capping enzyme forming a dodecameric ring [37]. HEV-3 and CHIKV nsP1 structures are overall similar and fold into three regions, referred to as the crown, waist, and skirt in CHIKV nsP1 (Figure 2A). However, these two proteins also present structural differences. The HEV-3 nsP1 starts with a 19 residue α-helix, which is absent in CHIKV nsP1. In contrast, the CHIKV nsP1 ends with a 9 residues α-helix, packed against two α-helices of the crown, which is absent in HEV-3 nsP1. Lastly, although the membrane-binding and oligomerization (MBO) loop 1 points in the same direction in both structures, the CHIKV MBO loop 2 is replaced by a α-helical motif in HEV-3, encompassing a long and two short α-helices (Figure 2A).

Since the CHIKV nsP1 structure revealed a dodecameric assembly, we then tested whether such oligomerization state might be compatible with the HEV-3 nsP1 predicted structure. However, with a total of 5400 residues, the structure prediction of a HEV-3 nsP1 dodecamer as a whole is beyond the possibilities of AF2 notebooks. Therefore, we generated the HEV-3 nsP1 dodecamer with SymmDock, a server for the prediction of complexes with Cn symmetry by geometry-based molecular docking [42]. The HEV-3 nsP1 dodecamer thus predicted did not show significant clashes at the subunit interfaces (Figure 2B). Moreover, we predicted a HEV-3 nsP1 dimeric assembly with AF2, which was compatible with nsP1 dimers within the dodecamer (Figure 2B, inset), besides a small rotation of ~4° between the two subunits. It is also noteworthy that the HEV-3 nsP1 N-terminal helix fits well in the dodecameric structure (Figure 2B). The HEV-3 nsP1 dodecamer has an outer diameter of ~170 Å as compared with 186 Å for CHIKV nsP1. Moreover, the HEV-3 nsP1 internal pore, with a diameter of 55 Å at the crown level and of 40 Å at the higher waist level, where loop 350–360 protrudes in the pore (Figure 2A,B), is also smaller than that of CHIKV nsP1 with an internal diameter of 70 Å. To note, a large chain break occurs at the same position in CHIKV nsP1 (residues 358–375), suggesting that the pore would also be much smaller in this region (Figure 2A).

### 3.3. HEV-3 nsP2: A Metal Binding Protein

The HEV-3 nsP2 boundaries are at residues 516 and 689 of the polyprotein pORF1. The Dali server returned a significant hit (rmsd of 1.1 Å and a very high Z-score of 29.6) with a HEV-1 nsP2 (residues 474–649) whose structure has been solved by X-ray crystallography [33]. HEV-1 nsP2 shares 70% sequence identity with that of HEV-3. HEV-1 nsP2 was shown to be similar to a fatty acid binding domain (FABD) and contains a Zn^2+^, coordinated by His 631 and Glu 633. However, the side chain of His 686, identified as a potential Zn^2+^ binder, is solvent exposed.

The HEV-3 nsP2 His 671 and Glu 673 residues have their side chains in a position similar to those of His 631 and Glu 633 of HEV-1 nsP2, despite the absence of the Zn^2+^ ion in the predicted structure (Figure 3). Moreover, the HEV-3 C-terminus is close to these Zn^2+^ coordinating residues His 671 and Glu 673, and the HEV-3 His 686 side chain occupies the volume of the Zn^2+^ ion in the HEV-1 nsP2 structure (Figure 3). Despite displaying a FABD fold, both HEV nsP2 do not possess an internal cavity that could accommodate a fatty acid. Instead, a pre-C terminal helix sits within the potential fatty-acid-binding site in both nsP2 structures.

### 3.4. HEV-3 nsP3 Is a Macro Domain

Boundaries of HEV-3 nsP3 are at residues 793 and 941 of the polyprotein pORF1. This protein can be identified as a macro domain mono-ADP-ribose (MAR) hydrolase, since a Dali search returned several significant hits with such enzymes. The best hit was obtained with the structure of the Tylonycteris bat coronavirus CoV-HKU4 macro domain in complex with ADP-ribose (Table 1) [38].

The Rossmann-like fold is well conserved in both proteins, although two additional helices are observed in the bat CoV-HKU4 macro domain structure (Figure 4A,B). Superimposition of both structures shows that the ADP-ribose binding site is well formed in the HEV-3 nsP3 predicted structure, in which a bridge-like region sits above the phosphate groups (Figure 4D,E). In this sense, the HEV ADP-ribose-protein hydrolase activity has been demonstrated, using a protein construction (residues 768–943) from genotype 1 [43].

### 3.5. HEV-3 nsP4: A Helicase

The HEV-3 nsP4 boundaries are at residues 944 and 1223 of the polyprotein pORF1. The predicted bi-lobed structure of nsP4, which shows a deep internal crevice (Figure 5A), returned several significant hits with viral helicases using Dali. The best hit (Table 1) was obtained with the crystal structure of CHIKV helicase domain in complex with ADP-AlF_4_ (an ATP analog) and a 14-mer ssRNA, of which 7 bases are visible in the electron density map [39]. The CHIKV helicase, belonging to the superfamily 1 (SF1) helicases, encompasses four domains: a unique and small N-terminal domain (NTD), the SF1 familiar accessory domain 1B, a Stalk a-helix, and the two Rec-A-like domains (RecA1 and RecA2 linked by a a-helical connector) (Figure 5B,C).

The HEV-3 nsP4 helicase is much more compact than that of CHIKV helicase: it lacks the NTD and the connector, and the stack helix is much shorter. The 1B domain surrounding the stalk helix is also absent (Figure 5C). However, we superimposed both helicases and checked whether the HEV-3 helicase, despite being smaller than that of CHIKV, could bind the same ligands as those of the CHIKV helicase (Figure 5D,E). The ADP-binding site is well formed in HEV-3 nsP4, even though the ADP moiety is less buried than in the CHIKV helicase due to the absence of the connector helix (Figure 5D). The ssRNA occupies the crevice between the two RecA domains, but it is more solvent-exposed since the stalk-surrounding domain is absent (Figure 5D). This structural analysis indicates that the HEV-3 helicase is likely functional. Moreover, these discrepancies between both viruses are not surprising since in CHIKV, the helicase domain is associated in C-terminal with the protease, to form the Alphavirus nsP2 bi-functional protein.

### 3.6. HEV-3 nsP5: The RNA-Dependent RNA Polymerase

Boundaries of HEV-3 nsp5 are at residues 1242 and 1700 of the polyprotein pORF1. The nsP5 predicted structure presents the canonical right-hand shape of all RNA-dependent RNA polymerases, including the fingers, palm and thumb subdomains [44] (Figure 6A,B).

Dali returned several significant hits with viral RdRps, among which the Classical Swine Fever Virus NS5B RNA polymerase RNA-dependent [40] gave the best statistics. However, we superimposed the HEV-3 nsP5 predicted structure to that of the enterovirus 71 RdRp, which provided good statistics with Dali (6lse; Z-value = 17.4; rmsd = 3.9; lali = 344/464), since it is bound to a dsRNA [45] (Figure 6B,C). Both RdRps are similar in size and folding (Figure 6C). Moreover, HEV-3 nsP5 also contains a positively-charged tunnel that could accommodate dsRNA in a similar way to what is observed with the enterovirus 71 RdRp (Figure 6D,E). The enterovirus 71 RdRp catalytic residues Asp 329 and Asp 330, which belong to the conserved RdRp motif C, superimpose very well to HEV-3 nsP5 residues Asp 1566 and Asp 1567, that are at ~4.0 Å from the growing RNA chain end (Figure 6F).

### 3.7. HEV-3 nsPs May Not Assemble Pre-Formed Replicative Complexes in the Absence of RNA Substrates

Structure predictions of the HEV-3 polyprotein segment 1 and segment 2, which mimic the polyprotein pre-cleavage state, did not reveal any interactions between the five domains (Figure 1 and Figure 7).

However, we wished to investigate whether this lack of interaction would also be obtained with the individual nsPs. To this end, we performed two structural predictions using nsP2 to nsP5 as inputs, excluding nsP1 that likely forms a dodecamering ring. AF2 did not predict contacts between the different nsPs, even for the helicase (nsP4) and the RdRp (nsP5) which are two viral activities well-known to cooperate in (+)RNA viruses [46,47]. However, this prediction was performed in the absence of ssRNA or dsRNA, as AF2 predictions are restricted to proteins, and we cannot exclude that protein-protein interactions may occur in the presence of RNA substrates.

## 4. Discussion

Hepatitis E virus was ranked 6th among viruses with high animal-to-human spillover potential, just behind viruses causing hemorrhagic fevers (e.g., Lassa and Ebola viruses) and the SARS-CoV-2 [48]. Therefore, there is an urgent need to increase our knowledge on HEV and, in particular, on its replication machinery, which is a target of choice for efficient antiviral treatments. HEV produces a polyprotein encompassing several enzymes involved in the replication/translation process. The boundaries of these enzymes were predicted three decades ago based on sequence analyses [8]. Here, we show that the HEV-3 polyprotein pORF1 structure predictions by AF2 result in an amino-acid chain encompassing five domains (or nonstructural proteins) interspaced by linkers of variable lengths. The well-folded regions, which can be ascribed to enzymatic domains, present high pLDDT values, whereas the poorly structured linkers between them have low pLDDT values. Moreover, function assignments to the predicted domains are supported by the identification of structural homologs with known function. Indeed, it is expected that a helicase, for example, shares structural homology with other helicases, at least in its core, whereas additional domains or surface loops may differ from one another.

Although AF2 can predict the structure of virtually any protein from its sequence, it is not able to include any ligand in the predictions, except proteinaceous ones. Therefore, the positions of protein-bound nucleic acids, ions or organic co-factors cannot directly result from AF2 predictions. Our structural predictions of HEV-3 enzymes forming the replication/transcription machinery led us to identify structural homologs for each of them in the PDB using the Dali server. Interestingly, the superimposition of the predicted structures to the best Dali hits encompassing nucleic acids or ions made it possible to identify the position of these elements in the predicted structures. A striking result was that the volumes for these ligands or substrates were available in the predicted structures that were in a “ready to bind” conformation.

The predicted HEV-3 nsP1 structure exhibits a fold highly resembling that of CHIKV nsP1, but with different secondary structure elements at the N- and C-termini and in one of the MBO loop. Despite this, HEV-3 nsP1 forms well-packed dodecamers, similar to the CHIKV capping pore for replication factory [37], without clashes between domains. The HEV-3 nsP2 predicted structure is very close to the HEV-1 nsP2 crystal structure (1.1 Å rmsd). The latter has also revealed a Zn^2+^-binding domain and a fatty-acid-binding domain [33]. The larger deviation between both structures occurs in the Zn^2+^-binding site (Figure 3C). Interestingly, this domain partially overlaps, in the N-terminal part, with the putative cysteine protease predicted by computational homology analysis [8]. Thus, it was proposed from the HEV-1 nsP2 crystal structure that the HEV protease activity may be regulated through the binding of a fatty acid and also through a metal ion as cofactors. In particular, the exit of the pre-C terminal helix (Figure 3A) might allow lipid binding and induce a C-terminus rearrangement, which would turn it into a catalytically active protease [33]. Another hypothesis has also been raised suggesting that instead of Zn, HEV nsP2 through its highly conserved cysteines could ligate iron–sulfur (Fe–S) cluster, thereby acting as a cofactor for protease activity [15,33]. Overall, Zn has long been known to replace endogenous and O_2_ labile Fe–S metal cofactors during standard aerobic purification of proteins. The key role of Fe–S clusters in viral enzyme activities is an emerging field [49,50] and deserves greater attention, especially to the HEV putative cysteine protease. However, these hypotheses have not been confirmed to date and the existence of a protease activity in HEV is still an open question. The macro (or X) domain is a highly conserved protein found in all kingdoms of life. It catalyzes the removal of ADP-ribose molecules from ADP-ribosylated proteins. This post-translational modification regulates a wide variety of cellular processes (for review see [51]). In (+)RNA viruses, in addition to *Hepeviridae*, the macro domain is also found in *Coronaviridae* and *Togaviridae*. Its exact role is not known yet, but it seems to be involved in the viral pathogenicity. The HEV ADP-ribose-protein hydrolase activity has been demonstrated [43]. The HEV-3 macro domain (nsP3) and helicase (nsP4) have both structural homologs in the PDB, but they are overall more compact as they lack surface loops or domain(s). A striking feature of these predicted structures is that their substrate-binding sites, which accommodate ADP-ribose molecules for nsP3 and ATP and ssRNA for nsP4, are similar to those observed in experimental structures of viral macro domains and helicases, regarding their position and architecture. The HEV RdRp structure presented here is the first example of a RdRp of the alphavirus-like superfamily and thus fills the last branch of RNA viruses for which no RdRp structure was available. The catalytic residues (GDD) of the HEV-3 RdRp (nsP5) predicted structure, belonging to the motif C of RdRp [44], superimpose well with those of other RdRp, and the RNA-binding tunnel is also conserved.

By analogy to Alphavirus, it was suggested that the domains of the HEV replication/transcription machinery may interact with each other in the polyprotein context, and/or in the context of individual nsPs produced after cleavage of the polyprotein pORF1. However, the AF2 structure predictions of the whole HEV-3 polyprotein pORF1 did not reveal any inter-domain interaction, a result that was also observed with structure predictions using nsP2, nsP3, nsP4 and nsP5 together as input. In contrast, our AF2 structure predictions of the well-known SARS-CoV-2 replication complex, assembling the RdRp nsp12 with its nsp7 and nsp8 co-factors, resulted in a protein complex comparable to that observed in experimental structures [52] (Appendix A). This result suggests that, at least in the absence of RNA, the different HEV-3 nsPs may not interact with each other.

AF2 structural predictions are a potent and fast tool to determine the structure–function relationships of viral replication/translation machineries. Precisely knowing domain boundaries is highly valuable to recombinantly produce stable and active proteins and perform structural and functional analyses [32]. In particular, predicted structures can be used as templates to design mutants and test their activity, or to design inhibitors of the replication machineries, which is a target of choice for efficient antiviral treatments as illustrated by the NS5B-RdRp inhibitor used to treat HCV-infected patients [53].

## Figures and Tables

**Figure 1 viruses-14-01537-f001:**
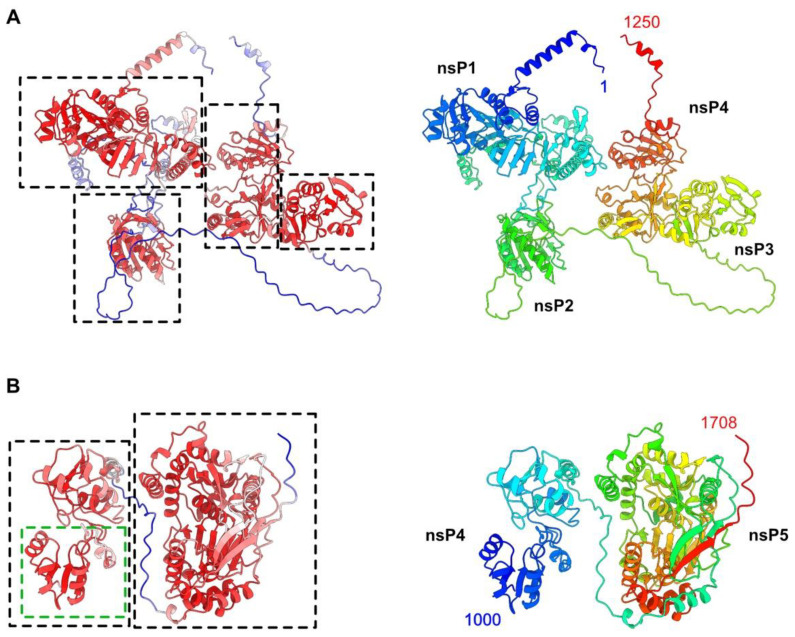
HEV-3 polyprotein pORF1 predicted structure. (**A**) Ribbon representation of the polyprotein segment 1 predicted structure (residues 1–1250), colored by pLDDT values (left) from red (best) to blue (worse), and rainbow colored from N-terminus to C-terminus (right) (see also Appendix A). The four domains are identified by dotted rectangles. The predicted HEV-3 nsPs are numbered according to their order in sequence. (**B**) Ribbon representation of the polyprotein segment 2 predicted structure (residues 1000–1708), colored by pLDDT values (left) from red (best) to blue (worse), and rainbow colored from N-terminus to C-terminus (right) (see also Appendix A). The C-terminal domain is indicated by a dotted rectangle. The first domain (green dotted rectangle) is part of nsP4 in segment 1.

**Figure 2 viruses-14-01537-f002:**
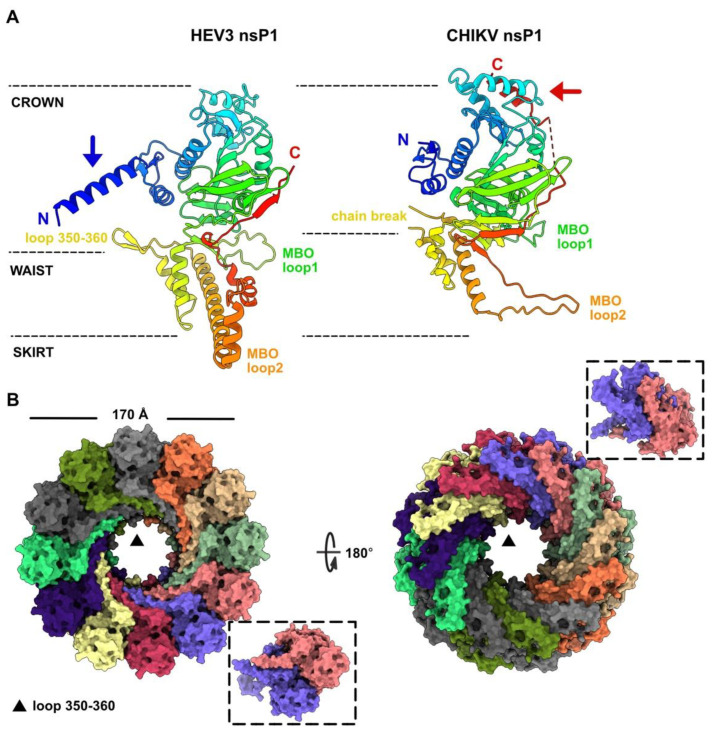
HEV-3 nsP1 predicted structure, a potential capping pore of the replication factory. (**A**) Ribbon representations of HEV-3 nsP1 predicted structure (left) and CHIKV nsP1 cryo-EM structure (right) (rainbow colored). The three nsP1 regions are identified with dashed lines, the membrane-binding and oligomerization (MBO) loops are labelled, and the N- and C-termini are indicated. The blue arrow indicates the position of the N-terminal α-helix in HEV-3 nsP1, and the red arrow indicates the position of the C-terminal α-helix in CHIKV nsP1. (**B**) Surface representation of the HEV-3 nsP1 dodecameric assembly (colored by chain) viewed from the crown (left) and from the MBO loop 2 (right). The insets highlight an AF2 predicted structure of a dimeric HEV3 nsP1.

**Figure 3 viruses-14-01537-f003:**
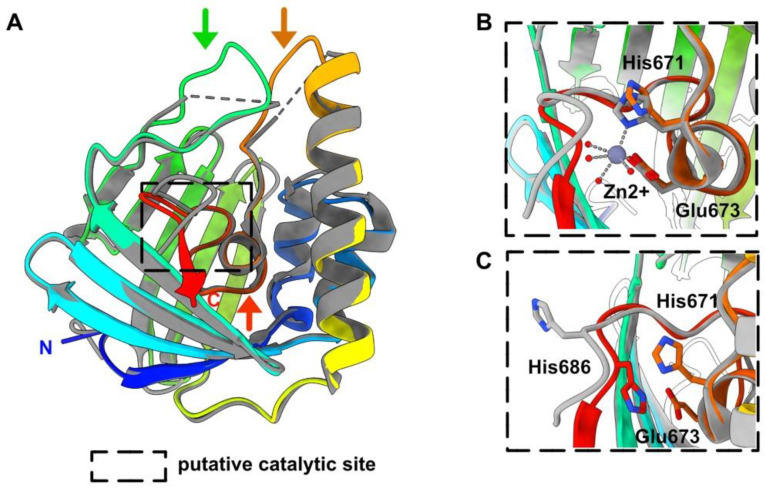
HEV-3 nsP2 predicted structure, a metal binding protein. (**A**) Ribbon representation of the HEV-3 nsP2 predicted structure (rainbow colored), superimposed to the HEV-1 nsP2 crystal structure (in grey) [33]. The green and orange arrows indicate the position of loops predicted by AF2 and not visible in the HEV-1 nsP2 crystal structure. The red arrow points to the position of the pre-C-terminal helix that would sit in the putative fatty acid-binding cavity in both nsP2 structures. (**B**,**C**). Insets on the putative catalytic site showing the Zn^2+^ ion, its coordinating residues His 671 and Glu 673, and the His 686 that move towards the center of the protein in the HEV-3 nsP2 predicted structure.

**Figure 4 viruses-14-01537-f004:**
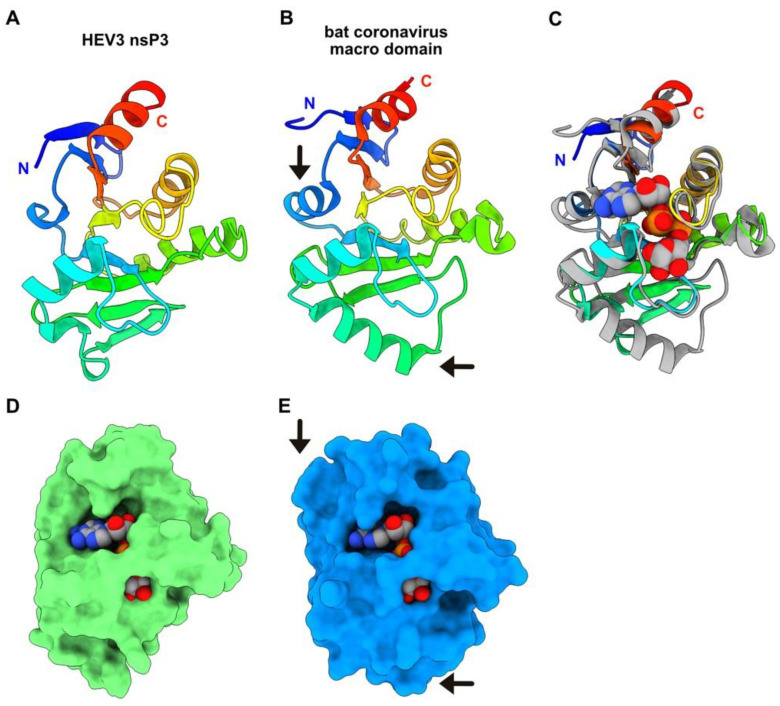
HEV-3 nsP3 predicted structure, a macro domain mono-ADP-ribose hydrolase. (**A**) Ribbon representation of the HEV-3 nsP3 predicted structure (rainbow). (**B**) Ribbon representation of the macro domain of bat CoV-HKU4 (rainbow colored). The arrows point to the position of the two additional helices as compared with HEV-3 nsP3. (**C**) Superimposition of HEV-3 nsP3 (rainbow) with bat CoV-HKU4 macro domain (grey). The ADP-ribose is atom colored and displayed as spheres (C: grey; O: red; N: blue; P: orange). (**D**) Surface representation of HEV-3 nsP3 with the ADP-ribose in the same position as in the bat CoV-HKU4 macro domain, after superimposition of both domains onto each other (panel C). (**E**) Surface representation of bat CoV-HKU4 macro domain in complex with ADP-ribose. The arrows point to the two extra helices as in panel B.

**Figure 5 viruses-14-01537-f005:**
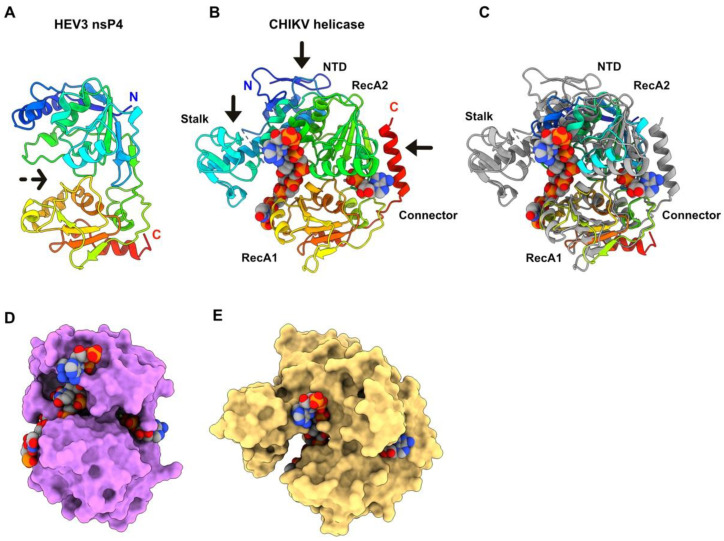
HEV-3 nsP4 predicted structure, a helicase. (**A**) Ribbon representation of HEV-3 nsP4 predicted structure (rainbow colored). The dotted arrow indicates the internal crevice. (**B**) Ribbon representation of CHIKV helicase with labelled domains. The arrows point to the parts missing in HEV-3 nsP4. The ADP-AlF_4_ and ssRNA molecules are atom colored and shown as spheres (C: grey; O: red; N: blue; P: orange). (**C**) Superimposition of HEV-3 nsP4 helicase (rainbow) and CHIKV helicase domain (grey). The ADP-AlF_4_ and ssRNA are atom colored and shown as spheres. (**D**) Surface representation of HEV-3 nsP4 with ADP-AlF_4_ and ssRNA in the same position as in the CHIKV helicase, after superimposition of HEV-3 nsP4 and CHIKV helicase onto each other (in C). (**E**) Surface representation of CHIKV helicase with bound ADP-AlF_4_ and ssRNA.

**Figure 6 viruses-14-01537-f006:**
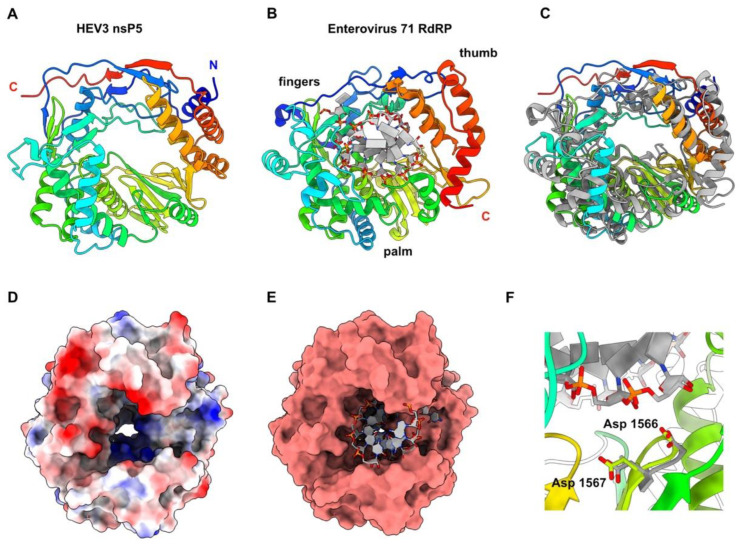
HEV-3 nsP5 predicted structure, the viral RdRp. (**A**) Ribbon representation of HEV-3 nsP5 predicted structure (rainbow colored). (**B**) Ribbon representation of the enterovirus71 RdRp (rainbow) in complex with dsRNA (colored by atoms, C: white; O: red; N: blue; P: orange). (**C**) Superimposition of HEV-3 nsP5 (rainbow) and the enterovius 71 RdRp (grey). (**D**) Surface representation of HEV-3 nsP5 colored by electrostatic potential. The internal tunnel is positively charged, in accordance with RNA synthesis. (**E**) Surface representation of HEV-3 nsP5 with dsRNA in the same position as in the enterovirus71 RdRP, after superimposition of both proteins onto each other (in C). (**F**) Highlight on HEV-3 nsP5 Asp 1566 and Asp 1567 and enterovirus71 RdRP catalytic residues Asp 329 and Asp 339.

**Figure 7 viruses-14-01537-f007:**
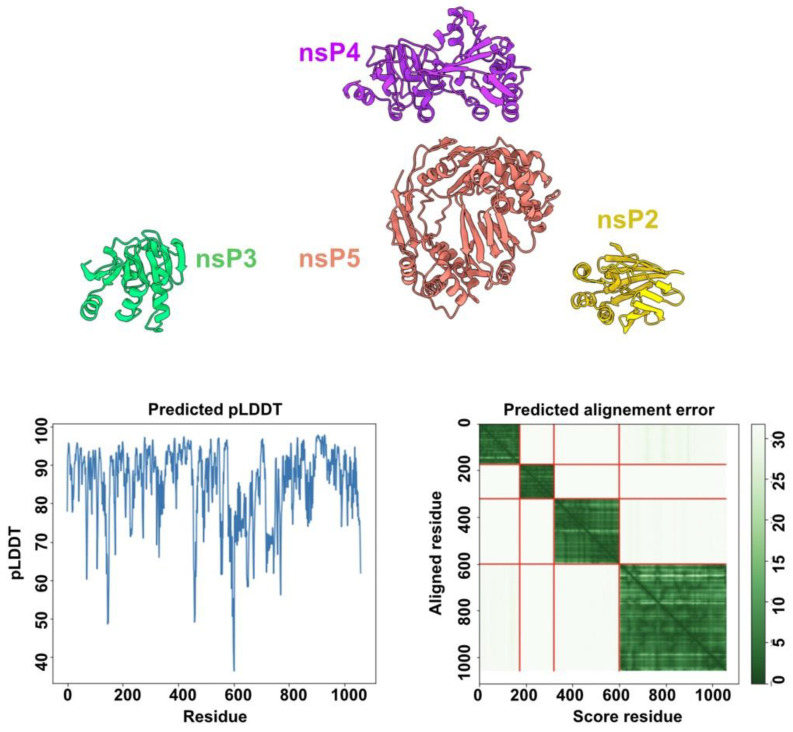
Structure prediction using the individual nsP2 to nsP5 as inputs. (**Top**) Ribbon representation of the four nsP predicted structures that do not contact with each other (the four nsP sequences were submitted together to AF2 for structure prediction). The prediction pLDDT scores (**bottom left**) and predicted alignment errors (**bottom right**) are shown. The predicted alignment error plot indicates the absence of contacts between the four nsPs.

**Table 1 viruses-14-01537-t001:** nsP boundaries and structural similarity with viral proteins deposited in the PDB. CHIKV, Chikungunya virus; ZBD, Zinc-binding domain; CoV, coronavirus; CSFV, classical swine fever virus; RdRp, RNA-dependent RNA polymerase.

nsP	Boundaries	PDB-ID	Z-Dali	Rmsd Å	Aligned Residues	Function	Organism	Reference
1	9–459	6z0v	20.2	3.8	347/452	capping pore	CHIKV	[37]
2	516–689	6nu9	29.6	1.1	166/168	ZBD	HEV	[33]
3	793–941	6mea	18.4	1.9	138/162	macro domain	bat CoV-HKU4	[38]
4	944–1223	6jim	21.4	3.2	262/455	helicase	CHIKV	[39]
5	1242–1700	5y6r	20.7	3.5	363/667	RdRp	CSFV	[40]

## Data Availability

Predicted structures coordinates (PDB format) will be available in the Appendix A.

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
