# Peer review of "Structure Prediction and Analysis of Hepatitis E Virus Non-Structural Proteins from the Replication and Transcription Machinery by AlphaFold2"

_viruses, 2022, doi:10.3390/v14071537_

Round 1
Reviewer 1 Report
Dear Authors,
The text titled "Structure prediction and analysis of Hepatitis E Virus non-structural proteins from the replication and transcription machinery by AlphaFold2" is well written and the authors explained accurately the structures and similarities with other viral protein. AlphaFold2 resulted a fast and potent tool to predict HEV pORF1 structure and can be use easily with other protein too predict structure and function.
Just one recommendation, in the text refer to genotypes as HEV-1, HEV-2, HEV-3 and HEV-4
Line 14: change "four types (HEV1-4)" in "four genotypes (HEV-1 to HEV-4)"
Line 51: change genotypes 3 and 4 with HEV-3 and HEV-4, as the same in the whole text
Line 106: chage gt3 with HEV-3
Reviewer 2 Report
Interesting paper looking at structure prediction of HEV ORF1, which is a long standing problem in HEV virology.
- The taxonomy section in the intro needs to be rewritten as per the latest ICTV taxonomy release on the family Hepeviridae.
- Should note in line 56 that Orthohepevirus C (new classification: Rocahepevirus ratti) can also cause chronic hepatitis in immunocompromised patients.
- Line 57: ‘sequenced’ is better word than isolated here.
- For non-expert readers, should include additional background on why AF2 was applied to HEV ORF1 structure prediction i.e. difficulties in obtaining actual structures using traditional methods requiring de novo structure prediction.
- Why was the Kernow C1 clone used for structure prediction when there are plenty of reference sequences of wild-type HEV available on GenBank? Kernow C1 is somewhat idiosyncratic in terms of having a human-ribosome derived ORF1 insert.
- How do the predicted domain boundaries compare to the classical computational predictions by Koonin et al?
- It is reassuring that the predicted nsp2 is similar to the corresponding HEV-1 crystal structure. For other nsps, is some form of validation possible? For example, homology modeling (e.g. I-TASSER) against alphavirus crystal structures should be theoretically possible for most nsps. RMSDs of these structures could then be compared to those independently derived from AF2. Would be interesting to see if the unique features uncovered in various domains by AF2 are replicated by homology modeling.
Round 2
Reviewer 2 Report
Addressed comments.